# [^18^F]2-Fluoro-2-deoxy-sorbitol PET Imaging for Quantitative Monitoring of Enhanced Blood-Brain Barrier Permeability Induced by Focused Ultrasound

**DOI:** 10.3390/pharmaceutics13111752

**Published:** 2021-10-20

**Authors:** Gaëlle Hugon, Sébastien Goutal, Ambre Dauba, Louise Breuil, Benoit Larrat, Alexandra Winkeler, Anthony Novell, Nicolas Tournier

**Affiliations:** 1CEA, CNRS, Inserm, BioMaps, Université Paris-Saclay, 91401 Orsay, France; gaelle.hugon@universite-paris-saclay.fr (G.H.); sebastien.goutal@universite-paris-saclay.fr (S.G.); ambre.dauba@universite-paris-saclay.fr (A.D.); louise.breuil@universite-paris-saclay.fr (L.B.); alexandra.winkeler@universite-paris-saclay.fr (A.W.); anthony.novell@universite-paris-saclay.fr (A.N.); 2CNRS, CEA, DRF/JOLIOT/NEUROSPIN/BAOBAB, Université Paris-Saclay, 91191 Gif-sur-Yvette, France; benoit.larrat@cea.fr

**Keywords:** blood-brain barrier, integrity marker, sorbitol, positron emission tomography, focused ultrasound

## Abstract

Focused ultrasound in combination with microbubbles (FUS) provides an effective means to locally enhance the delivery of therapeutics to the brain. Translational and quantitative imaging techniques are needed to noninvasively monitor and optimize the impact of FUS on blood-brain barrier (BBB) permeability in vivo. Positron-emission tomography (PET) imaging using [^18^F]2-fluoro-2-deoxy-sorbitol ([^18^F]FDS) was evaluated as a small-molecule (paracellular) marker of blood-brain barrier (BBB) integrity. [^18^F]FDS was straightforwardly produced from chemical reduction of commercial [^18^F]2-deoxy-2-fluoro-D-glucose. [^18^F]FDS and the invasive BBB integrity marker Evan’s blue (EB) were i.v. injected in mice after an optimized FUS protocol designed to generate controlled hemispheric BBB disruption. Quantitative determination of the impact of FUS on the BBB permeability was determined using kinetic modeling. A 2.2 ± 0.5-fold higher PET signal (*n* = 5; *p* < 0.01) was obtained in the sonicated hemisphere and colocalized with EB staining observed post mortem. FUS significantly increased the blood-to-brain distribution of [^18^F]FDS by 2.4 ± 0.8-fold (*V*_T_; *p* < 0.01). Low variability (=10.1%) of *V*_T_ values in the sonicated hemisphere suggests reproducibility of the estimation of BBB permeability and FUS method. [^18^F]FDS PET provides a readily available, sensitive and reproducible marker of BBB permeability to noninvasively monitor the extent of BBB disruption induced by FUS in vivo.

## 1. Introduction

The blood-brain barrier (BBB) plays a critical role in protecting the brain from potentially harmful substances of the circulation while controlling brain homeostasis [1]. Integrity of the BBB is mainly carried out by tight junctions between adjacent endothelial cells forming the brain microvasculature [2]. This considerably limits the paracellular (i.e., between cells) passage of solutes across the BBB which contributes to the sanctuary site property of the brain [2,3]. Many drug molecules and therapeutics cannot naturally permeate the intact BBB into the brain parenchyma. As a consequence, the BBB is a bottleneck in the development of central nervous system (CNS)-targeting therapeutics, which complicates the treatment of brain diseases, notably cerebral malignancies [4].

Numerous strategies and technologies have been proposed to overcome the BBB and achieve sufficient brain delivery of therapeutics [5]. Among these strategies, focused ultrasound in combination with microbubbles (FUS) is emerging as an effective means to locally and temporarily enhance BBB permeability, mainly through the “opening” of the paracellular route by disruption of the tight junctions [6]. A large body of preclinical research has convincingly shown that FUS enhanced the brain exposure of small molecules, biologics or gene therapy [7,8,9,10,11]. More than 30 clinical trials using FUS-induced BBB disruption are ongoing and assessed the translational potential and clinical safety of the technique [12,13,14,15,16,17].

The development of FUS is tightly linked to the availability of markers of BBB integrity, with the aim to quantitatively assess the impact of various FUS conditions in enhancing BBB permeability [18]. In the literature, many different BBB integrity markers have been described for nonclinical in vitro and ex vivo (terminal) experiments [19]. Low molecular weight (MW, g/mol) hydrophilic molecules such as fluorescein (MW = 332, fluorescent detection), (radio)labeled analogues of sucrose (MW = 342) or mannitol (MW = 182) (Figure 1A) are usually preferred as quantitative “paracellular” markers of membrane integrity (Figure 1) [1,20]. Low MW is associated with enhanced sensitivity to subtle change in barrier permeability compared with higher MW compounds such as Evan’s blue (EB, MW = 961, highly bound to plasma proteins), radiolabeled proteins (albumin MW = 65,000), dextrans (MW = 1500–70,000) or inulin (MW = 6179) [1,21].

Translational imaging techniques such as magnetic resonance imaging (MRI), single-photon emitting computed tomography (SPECT) or positron emission tomography imaging (PET) are increasingly regarded as methods to noninvasively monitor the impact of FUS on BBB permeability [18]. So far, dynamic contrast-enhanced (DCE), T1 relaxometry and dynamic susceptibility contrast MRI using gadolinium (Gd)-based contrast agents such as gadoterate (MW = 558), Gd-DTPA (Gd-diethylene triamine pentaacetic acid, MW = 546) [22,23] or large iron nanoparticles [24] have been used to detect BBB disruption in vivo. Brain SPECT using [^99m^Tc]DTPA (MW = 487) is also widely used as a BBB integrity marker [25]. The predominant use of MRI and SPECT to investigate BBB integrity is likely due to the wide availability of corresponding imaging probes [26]. In humans, the clinical proof-of-concept of FUS-induced BBB disruption has been achieved using DCE-MRI to assess and localize BBB disruption in vivo [12]. However, accurate and absolute quantification of the brain penetration of contrast agent using MRI is very challenging [22]. More quantitative neuroimaging techniques are needed to accurately estimate the impact of FUS on BBB integrity.

Compared with MRI or SPECT, PET imaging benefits from absolute quantitative performances and high sensitivity so that quantitative determination of the concentration of microdose radiolabeled compounds in the brain is possible. Furthermore, high frame rate allows for kinetic modeling and estimation of transfer rate across the BBB [18]. For most imaging centers, the main limitation of PET is the limited availability of compounds other than commercial and daily produced radiopharmaceuticals such as [^18^F]2-fluoro-2-deoxy-D-glucose ([^18^F]FDG, Figure 1). Consistently, PET using radiolabeled low- or high-MW makers of BBB integrity such as [^18^F]1-fluoro-1-deoxy-D-mannitol (MW = 183) [27] or [^11^C]inulin (MW = 6179) [28], which synthesis requires production of radioisotope by a cyclotron and dedicated radiochemistry facilities, did not reach mainstream use.

Sorbitol (Figure 1B), a stereoisomer of mannitol, is a non-transported hydrophilic small molecule (MW = 182) that is poorly metabolized in mammals. The fluorinated derivative [^18^F]2-fluoro-2-deoxy-sorbitol ([^18^F]FDS, MW = 183, Figure 1C) can be virtually obtained in all nuclear medicine departments and molecular imaging laboratory from simple chemical reduction of commercial [^18^F]FDG [29]. [^18^F]FDS shows nonsignificant brain uptake in healthy rodents and humans [30] and benefits from favorable pharmacokinetic properties for quantitative PET imaging in vivo [31].

This study aimed at evaluating [^18^F]FDS PET for quantitative monitoring of enhanced BBB permeability induced by FUS in vivo. To this end, transcranial FUS conditions were optimized to induce hemispheric BBB disruption in mice. The kinetics of [^18^F]FDS across the BBB were compared in the sonicated and nonsonicated hemispheres.

## 2. Materials and Methods

### 2.1. Production of [^18^F]FDS

Synthesis of [^18^F]2-fluoro-2-deoxy-sorbitol ([^18^F]FDS) from commercial [^18^F]FDG (Figure 1C) and quality control was described by Li et al. [29]. Briefly, NaBH_4_ (2 mg) was added to 4 mL [^18^F]FDG (180–200 MBq, Curium, Saclay, France). After 15 min reaction at room temperature, 10 µL acetic acid (0.15 mmol) was added. The mixture was passed through a Sep-Pak Alumina-N-Plus-Long cartridge (Waters, Guyancourt, France). Radiochemical purity was checked using radio-thin-layer chromatography (TLC) using silica gel-coated alumina plates (Merck, Guyancourt, France). The mobile phase consisted in acetonitrile/water (80/20, *v/v*).

### 2.2. Focused Ultrasound

The method for spatially controlled BBB disruption was optimized from previous work [10] to induce reproducible BBB disruption in the right brain hemisphere only. Experiments were performed using female NMRI nu/nu mice. Seven-week-old mice were anesthetized with 1.5% isoflurane in O_2_/air (50/50, *v*/*v*). A catheter was inserted in the tail vein and the animal was transferred to the sonication system. As NMRI nu/nu mice lack body hairs shaving of the head could be omitted. Microbubbles (50 µL, SonoVue^®^, Bracco, Italy) were intravenously administrated in the tail vein before the beginning of the FUS (*n* = 5) or sham (no FUS, *n* = 3) session.

FUS were delivered by a spherically focused transducer (active diameter 25 mm, focal depth 20 mm, axial resolution 5 mm, lateral resolution 1 mm, Imasonic, Voray sur l’Ognon, France) centered at 1.5 MHz. The transducer was connected to a single-channel programmable generator (Image Guided Therapy, Pessac, France), mounted on a motorized *XYZ*-axis stage, and positioned above the mouse head maintained under anesthesia. The device was coupled to the mouse head using a latex balloon (filled with deionized and degassed water) and coupling gel. The distance between the transducer and the skull was adjusted by the displacement of the motorized axis (*Z*) and the filling of the balloon in order to target the center of the right brain hemisphere, at the focal distance (i.e., 20 mm). The FUS sequence used a peak negative pressure of 525 kPa (calibrated in deionized water). Therefore, the transmitted in situ pressure in the mouse brain was previously estimated to be 420 kPa considering a transmission loss through the skull of 20% at 1.5 MHz [32]. A mechanical scan (*XY*-axis) was synchronized to the generator output in order to induce a hemispheric brain BBB opening of 6 mm (anterior-posterior) × 3.6 mm (lateral right hemisphere). This 3.5 s sequence was repeated 36 times for a total exposure of 126.4 s with a global ultrasound duty of 71%.

### 2.3. Evan’s Blue Extravasation Test

Evan’s blue (EB) extravasation test was used as a positive control to visually check and localize BBB disruption induced by the FUS protocol. Solution of EB (obtained from Sigma-Aldrich, Saint-Quentin Fallavier, France) was freshly prepared at 4% in NaCl 0.9% as previously described [33]. Mice received 100 µL EB i.v., immediately after FUS. One hour after injection, i.e., at the end of PET acquisition, animals were euthanized and brains were removed to visually assess EB extravasation. Due to circulating radioactivity in [^18^F]FDS-injected animals, no perfusion washout was performed to remove blood and EB from the brain vasculature. Coregistration of the brain distribution of EB and the [^18^F]FDS PET signal in the brain obtained in an animal of the FUS group was performed. The frozen brain was sectioned with a cryostat (Leica CM3050 S, Leica, Wetzlar, Germany). Brain sections were scanned with a 20× lens, using an AxiObserver Z1 microscope (Zeiss, Jena, Germany) to observe distribution of EB-associated fluorescence in coronal slices. Then, representative EB brain slice and corresponding PET images obtained in the same animal were coregistered on MRI template to compare the PET signal in vivo with EB fluorescence ex vivo.

### 2.4. [^18^F]FDS PET Imaging

Immediately after EB injection (<60 s), anesthetized mice were transferred to the microPET scanner (Inveon, microPET, Siemens Healthcare, Knoxville, TN, USA). [^18^F]FDS was administered intravenously (4.2 ± 0.76 MBq) using a microinjection pump at the rate of 0.2 mL·min^−1^ during 60 s (*n* = 5 FUS; *n* = 3 sham). Dynamic PET acquisition (60 min) started with [^18^F]FDS injection.

PET images were reconstructed using the three-dimensional ordered subset expectation maximization with maximum a posteriori algorithm (3D OSEM/MAP) and corrected for attenuation, random coincidences and scattering. Volumes of interests (VOI_s_) were manually delineated using Pmod software (version 3.8, PMOD Technologies Ltd., Zurich, Switzerland). In the FUS group, extravasation of ^18^F-FDS was obvious in the sonicated area on late PET images. The region with disrupted BBB was delineated and mirrored to the contralateral hemisphere. In sham animals, VOI_s_ were drawn in each brain hemisphere. Another VOI was drawn on the aorta (blood-pool), obvious on early time-frames, to generate an image derived input function (IDIF).

Time activity curves (TACs) were corrected for radioactive decay, injected dose and animal weight and reported as standardized uptake value (SUV) vs. time. Area under the TAC (AUC) was calculated from either 0 to 15, 30 or 60 min (AUC_brain_) as well as corresponding AUC_brain_/AUC_blood_. Kinetic modeling of the 60 min PET data was performed using either the Logan graphical method [34] and a 1-tissue compartment (1-TC) model using IDIF to estimate the total volume of distribution (*V*_T_, mL·cm^−3^) of [^18^F]FDS and describe its transport across the BBB. Parametric images were generated to visualize the distribution of *V*_T_ (1-TC model) of [^18^F]FDS in the brain and peripheral tissues.

### 2.5. Statistics

Data are presented as mean ± S.D. Statistical analysis was performed using GraphPad Prism software version 9.1, La Jolla, CA, USA. Outcome parameters of kinetic modeling were compared using a two-way ANOVA with “hemisphere” and “group” as factors followed by Tukey’s multiple comparison test. Ratio of radioactivity measured in the right to the left hemisphere, as well as blood data obtained in the FUS and the sham group were compared using the Mann–Whitney U test. The Pearson’s test was used to test the correlation between different kinetic parameters obtained in the same brain regions. Statistical significance was set a *p* < 0.05. Intragroup variability of outcome parameters was estimated by the coefficient of variation (CV = 100 × S.D./mean).

## 3. Results

Chemical transformation of [^18^F]FDG into [^18^F]FDS was very effective (Figure 1). Radio-TLC analysis of the preparation revealed a single peak with retardation factor (Rf) = 0.6, consistent with the presence of [^18^F]FDS [29]. The Rf of untransformed [^18^F]FDG, tested using the same TLC conditions, was 0.9 (data not shown). No [^18^F]FDG could be detected using this method in the final preparation of [^18^F]FDS.

In FUS animals, strong EB staining was observed in the posterior part of the sonicated (right) hemisphere only (Figure 2A). This contrasted with the low EB staining still observed in brain vessels of the contralateral hemisphere, as well as in sham animals, because no washing out was performed to remove the circulating blood from the resected brain tissue. This confirmed that enhanced EB staining could be used as a positive control for effective BBB disruption.

Similarly, no washing out of circulating blood radioactivity was performed to interpret in vivo PET images. PET images in sham animals confirmed the negligible baseline brain uptake of brain [^18^F]FDS across the intact BBB (Figure 2B), whereas a strong PET signal could be monitored in the sonicated brain hemisphere of FUS animals. Parametric PET images, expressed in *V*_T-1TC;60min_ closely resembled the PET images in SUV units (Figure 2C). Images revealed a region with obvious increase in [^18^F]FDS uptake that was strictly localized in the right hemisphere, with very limited impact on the left hemisphere. Brain distribution of [^18^F]FDS PET signal was consistent with EB extravasation observed in the whole brain (Figure 2A).

Moreover, brain distribution of the fluorescent signal of EB observed ex vivo in the right brain hemisphere with higher spatial resolution was similar to the [^18^F]FDS PET signal obtained in the same animal of the FUS group (Figure 2D). It should be noted that although no washout was performed, no fluorescent signal was observed in the contralateral brain hemisphere underlining specific leakage of EB into the brain parenchyma. Interestingly, mapping of [^18^F]FDS brain distribution within the sonicated area displayed a gradient from the center to the periphery which was not observed using EB as a BBB integrity marker.

In hemispheres with intact BBB (sham animals or contralateral hemisphere), brain PET signal increased rapidly with maximal uptake at T_max_ = 4 min, followed by slow decrease of the radioactivity. FUS selectively enhanced the brain PET signal and T_max_ was achieved later at ~13.5 min, followed by a plateau (Figure 3A,B). SUV values in the sonicated (right) hemisphere at 15, 30 and 60 min were significantly higher than in the nonsonicated (left) hemisphere (*p* < 0.01, Figure 3A). Brain exposure (AUC_0–60min_) of [^18^F]FDS in the sonicated volume was 2.2 ± 0.5-fold higher compared with the contralateral volume (*p* < 0.01) and 1.8 ± 0.4-fold higher than the mean AUC_0–60min_ measured in the corresponding (right) hemisphere of the sham group (*p* < 0.05, Figure 4A). The ratio of the PET signal in the sonicated volume to the contralateral region increased from 0 to 15 min to reach a plateau (Figure 3C). FUS did not impact the kinetics [^18^F]FDS in the blood-pool with significant difference in neither SUV values nor AUC_blood_ between the FUS and the sham group (*p* > 0.05, Figure 3D).

Brain TACs of [^18^F]FDS were accurately described with acceptable fit (error < 3%) by *V*_T_ estimated using the 1-TC model (Figure 4B). The Logan graphical method provided similar estimation of brain *V*_T_ (correlation *p* < 0.001, R^2^ = 0.99, Figure 5). A significant increase in the *V*_T-1TC_ of [^18^F]FDS was observed in the sonicated region compared with the contralateral region (2.43 ± 0.8-fold increase, *p* < 0.01) or the brain of sham animals (1.9 ± 0.2-fold increase, *p* < 0.05) (Figure 4). In the FUS group, the variability of *V*_T-1TC_ values was lower in the sonicated brain (CV = 10.1%) compared with the nonsonicated brain (CV = 29.7%). The influx and efflux transfer rates K_1_ and k_2_ were estimated with acceptable fit (error < 7%). However, K_1_ values were associated with a high intragroup variability in the nonsonicated brain (CV = 37.6%) compared with the sonicated brain (CV = 18.5%), which questions the relevance of K_1_ and k_2_ to describe the extremely low BBB penetration of [^18^F]FDS across the intact BBB. *V*_T-1TC_ was therefore retained as the gold-standard parameter to describe the brain distribution of [^18^F]FDS in all tested conditions.

Other quantification methods were tested to either reduce the length of PET acquisition or simplify analysis of the PET data. *V*_T-1TC_ values estimated from 0 to 30 min (*V*_T-1TC;30min_), as well as AUC_brain_/AUC_blood_ measured from either 0 to either 30 or 60 min, was significantly correlated with *V*_T-1TC;60min_ (*p* < 0.001, R^2^ = 0.99) with limited loss in the sensitivity to detect enhanced brain distribution of [^18^F]FDS induced by FUS (Figure 5). However, further reduction of the length of PET frame tended to underestimate the brain distribution of [^18^F]FDS, especially in in the sonicated hemisphere. Nevertheless, AUC_brain_/AUC_blood_ from 0 to 15 min was still significantly correlated with V_T-1TC_ values (*p* < 0.001, R^2^ = 0.94) but the slope of the linear regression was <0.5. (Figure 5)

## 4. Discussion

In the present study, [^18^F]FDS PET imaging was validated for the first time as a translational and quantitative marker of BBB permeability to estimate the impact of spatially-controlled FUS on BBB integrity in mice.

Unlike [^18^F]FDG, [^18^F]FDS is poorly taken up by mammalian cells because it lacks transporters and/or enzymes for specific cell uptake and retention [29]. [^18^F]FDS PET has been previously validated in animals and humans to study renal function [35] or detect/estimate bacterial burden in tissues because sorbitol is a specific metabolic substrate of some strains of Gram-negative bacteria [36]. [^18^F]FDS therefore benefits from extensive clinical pharmacology and safety data as an experimental radiopharmaceutical drug [30]. However, to our knowledge, the impact of BBB integrity on the brain kinetics of [^18^F]FDS has never been evaluated so far.

[^18^F]FDS benefits from the characteristics of an “ideal” marker of BBB integrity [21]. This includes, metabolic stability, low binding to plasma proteins (<0.1%) and low baseline brain uptake when the BBB is intact [30,31]. Interestingly, Li et al. reported that [^18^F]FDS visually accumulated in orthotopic brain tumor xenografts in mice, despite negligible uptake by implanted glioma cells in vitro [29]. Given the substantial sensitivity of [^18^F]FDS to the integrity of the BBB, it can be hypothesized that the local BBB leakage in the tumor environment, rather than uptake by cancer cells, may explain the enhanced [^18^F]FDS PET signal observed by Li and colleagues in the tumor area [29]. Further experiments are however needed to validate the use of [^18^F]FDS to quantitatively monitor the permeability of the blood-tumor barrier. Similarly, we advocate that the high sensitivity of [^18^F]FDS to BBB integrity may be considered for correct estimation of bacterial burden in brain lesions using [^18^F]FDS PET [37].

Pharmacokinetic modeling of brain [^18^F]FDS PET data is relatively simple. Estimation of brain *V*_T_, using either the graphical method (Logan plot) or compartmental modeling (1-TC) provided similar outcome parameters to describe the distribution of [^18^F]FDS in both the sonicated and the non-sonicated brain, with low intragroup variability. In the FUS group, variability was strikingly lower (CV ~10%) in the sonicated brain regions compared with regions with the intact BBB (CV ~30%). This may be explained by the lower PET signal in the brain with intact BBB, associated with higher signal-to-noise ratio compared with the sonicated brain in which the PET signal is higher. Altogether, this suggests that *V*_T_, which takes any change in the plasma kinetics of [^18^F]FDS into account, provides a reliable and sensitive outcome parameter to describe the BBB permeation of [^18^F]FDS across the BBB. Estimation of *V*_T-1TC_ from 30 min PET data offers a compromise to observe and delineate the blood pool in aorta at early time-frames and allow for correct estimation of the brain distribution of [^18^F]FDS while reducing the total length of PET acquisition (Figure 5).

Simplified parameters were tested to quantitatively estimate the impact of FUS on the brain uptake of [^18^F]FDS. AUC_brain_/AUC_blood_ measured from 0 to 30 or 60 min after injection accurately predicted *V*_T_ and can be used as a surrogate parameter to estimate BBB permeation with limited impact on the sensitivity of the method compared with 60 min scan (Figure 5). *V*_T_ or AUC_brain_/AUC_blood_ should be used, rather than AUC_brain_, in situations where change in the blood kinetics of [^18^F]FDS is expected, such as renal disorders to correctly estimate the blood to brain distribution of [^18^F]FDS [38].

The FUS method reported in this project allows for localized BBB disruption in the right hemisphere while keeping the contralateral hemisphere intact, as confirmed using the post mortem EB extravasation test. In this situation, the contralateral hemisphere can serve as a convenient reference region to accurately estimate the impact of FUS on enhanced [^18^F]FDS brain uptake (AUC_brain_, Figure 3C). However, in many situations, BBB disruption induced by FUS or other methods may affect the whole brain or cannot be localized a priori [39]. In the absence of such reference region, increase in BBB permeability can be quantified by comparing [^18^F]FDS *V*_T_ or AUC_brain_/AUC_blood_ obtained in a control (sham) group or in the same animals before any intervention, following a longitudinal design. This illustrates the added value of absolute quantitative PET compared with other neuroimaging techniques for noninvasive determination of different levels of BBB permeation in various situations [26].

Mapping of enhanced [^18^F]FDS PET signal induced by FUS in the brain was consistent with EB extravasation observed ex vivo and in vitro, although exact co-registration of PET images with brain slices is very challenging. This suggests a limited diffusion of [^18^F]FDS from the sonicated to the nonsonicated brain with intact BBB. Interestingly, brain distribution within the sonicated area displayed a gradient from the center to the periphery (Figure 2). It may be hypothesized that heterogeneity in intensity of delivered ultrasound may occur within the sonicated area as a consequence of loss of ultrasound transmission relative to the transducer angulation at the skull surface. Interestingly, such a phenomenon could not be detected using high-MW markers of BBB integrity such as gadoterate or using EB extravasation in the same animals (Figure 2).

Several recent examples illustrate the added value of PET imaging to investigate the impact of FUS on BBB structure and function in vivo [18]. First, brain uptake of [^18^F]FDG, which is actively transported by glucose transporter 1 (GLUT1), was shown to be lower during FUS-induced BBB disruption, consistent with the local decrease in GLUT1 expression in the sonicated brain. Baseline [^18^F]FDG uptake and GLUT1 expression were fully restored 24 h after FUS [40]. P-glycoprotein (P-gp) and breast cancer resistance protein (BCRP) are the main efflux transporters expressed at the BBB [2]. We have shown that FUS-induced BBB disruption did not significantly impact the efflux transport of the P-gp substrate [^11^C]*N*-desmethyl-loperamide and the dual P-gp/BCRP substrate [^11^C]erlotinib in rats [33]. These particular PET probes may however lack the sensitivity to detect subtle changes in transporter function [41]. Decrease in P-gp expression has been reported 24–48 h after FUS [42,43]. However, long-term impact on efflux transport function at the BBB remains to be investigated in details. In this framework, [^18^F]FDS may offer a quantitative tool to estimate the dynamics of BBB disruption and restoration after FUS with suitable temporal resolution [24]. Altogether, this suggests that [^18^F]FDS enriches the PET imaging toolbox to investigate the short- and long-term impact of FUS on BBB permeability with respect to its molecular environment, including regulation of tight junction expression [18].

Simple production of [^18^F]FDS from commercial [^18^F]FDG makes it an appealing radiopharmaceutical candidate for determination of BBB permeability using quantitative PET. [^18^F]FDS is not a ”ready-to-inject” preparation compared with [^18^F]FDG, although radiopharmaceutical production and shipment could theoretically be ensured by an external manufacturer. However, most radiopharmaceuticals used in nuclear medicine and molecular imaging departments require handling by a radiopharmacy team, chemical reaction and quality control [44]. Efforts are moreover being made to develop kit formulation for radiosynthesis of [^18^F]FDS to simplify the production of this radiopharmaceutical [45]. Radiosynthesis of [^18^F]FDS for animal or clinical use can therefore be achieved in most molecular imaging departments, including those not equipped with radiochemistry facilities [37]. From a clinical perspective, the radiation dosimetry of this radiopharmaceutical was shown suitable for human use, although radiation exposure can be a limitation for the repeated use of [^18^F]FDS PET in humans [30].

This study has some limitations. The evaluation of the impact of FUS using [^18^F]FDS PET was evaluated in healthy animals only. As a consequence, FUS induced a large increase in the BBB penetration of [^18^F]FDS, starting from a low baseline brain uptake. In this situation, statistical significance was reached using a small number of individuals. However, a large body of translational research supports that BBB integrity is compromised in many CNS pathological conditions including multiple sclerosis, hypoxic/ischemic insult, traumatic injury, Parkinson’s and Alzheimer’s diseases, epilepsy and brain tumors [1]. Different non-clinical models have shown that BBB leakage is a common feature of the neuroinflammatory cascade and contributes to disease-associated brain damage [3,46]. PET imaging is expected to offer more quantitative insight into BBB permeability in such pathophysiological conditions [27,47,48,49]. Nevertheless, further experiments are needed, probably in a larger number of animals, to test whether [^18^F]FDS has the sensitivity and appropriate test-retest variability to be useful as permeability marker to investigate the dynamics of disease-associated changes in BBB integrity in longitudinal studies. NMRI nu/nu mice are widely used as a model for cancer treatment, including orthotopic brain tumors, which will be the logical next step for the evaluation of the impact of FUS using [^18^F]FDS PET imaging [50]. We assume that the use of female individuals of this immunodeficient mouse strain in the present study is not likely to bias the immediate impact of FUS-induced BBB disruption compared with male and/or immunocompetent mouse strains. Nevertheless, immunodeficiency was shown to improve the long-term integrity of the BBB in a model of intracerebral hemorrhage [51]. Longitudinal studies using quantitative [^18^F]FDS PET imaging may be useful to further investigate the time-dependent cross-talk between the immune system and BBB permeability in pathophysiological situations [46].

Restoration of BBB structure and function is increasingly the subject of investigations as a therapeutic target to prevent or limit the outcome of neurological disorders [52]. Conversely, temporary FUS-induced BBB disruption is also regarded as a potential therapeutic strategy in various CNS diseases. FUS alone was shown to achieve promising results at reducing the amyloid load in Alzheimer’s disease or at inducing neurogenesis [53,54]. This questions the long-term impact of the temporary FUS-induced BBB disruption in a situation where BBB integrity and function are already compromised [3]. This complex situation illustrates the crucial need for quantitative PET imaging techniques to untangle the impact of FUS on BBB permeability in relation to brain function in animal models and patients [18].

## 5. Conclusions

[^18^F]FDS PET imaging presents essential properties to become an effective and quantitative marker of BBB permeability, which include availability, safety, low MW, low distribution across the intact BBB and low diffusion from the sonicated volume to the non-sonicated brain with intact BBB. This is demonstrated here for the first time using local induction of BBB permeability by FUS. [^18^F]FDS PET offers a very instrumental imaging marker to finely and dynamically measure BBB permeability variations over time scales of minutes.

## Figures and Tables

**Figure 1 pharmaceutics-13-01752-f001:**
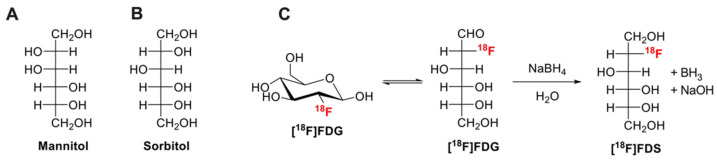
Chemical structures of mannitol (**A**), sorbitol (**B**). Chemical reduction of [^18^F]2-fluoro-2-deoxy-glucose ([^18^F]FDG) to [^18^F]2-fluoro-2-deoxy-sorbitol ([^18^F]FDS) using sodium borohydride (NaBH_4_) is shown in (**C**).

**Figure 2 pharmaceutics-13-01752-f002:**
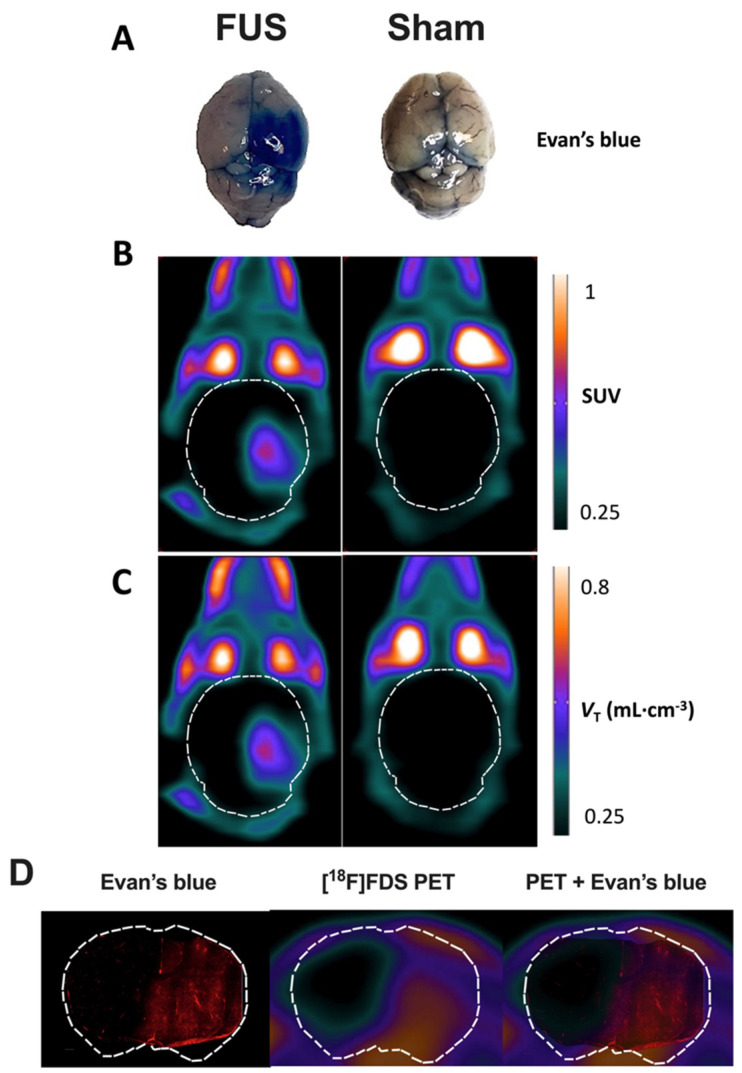
Impact of focused ultrasound (FUS) on blood-brain barrier (BBB) integrity assessed using Evan’s blue extravasation and positron emission tomography (PET) using [^18^F]2-fluoro-2-deoxy-sorbitol ([^18^F]FDS). Extravasation of Evan’s blue was obvious in the right brain hemisphere of animals of the FUS group (**A**). Standardized uptake value (SUV)-normalized brain PET images (sum 30–60 min) of [^18^F]FDS uptake after hemispheric BBB disruption induced by focused ultrasound (FUS) or without (sham) are shown in (**B**). Corresponding parametric images describing the total volume of distribution (*V*_T_), estimated using the one-tissue compartment model from 0 to 60 min are shown in (**C**). In (**D**), the distribution of the fluorescence of the Evan’s blue dye in a coronal slice of mouse brain after FUS protocol obtained ex vivo was overlaid to the corresponding slice of [^18^F]FDS PET image obtained in vivo in the same animal.

**Figure 3 pharmaceutics-13-01752-f003:**
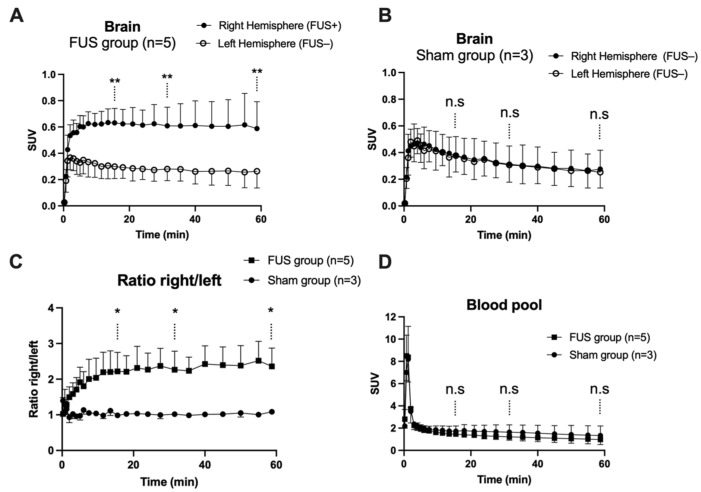
Brain kinetics of [^18^F]2-fluoro-2-deoxy-sorbitol ([^18^F]FDS). BBB disruption was obtained in the right brain hemisphere using focused ultrasound (FUS group, *n* = 5, in (**A**), but not in the sham group (*n* = 3, in (**B**)). The time course of the right/left ratio of the PET signal in FUS and sham animals is shown in (**C**). Corresponding image-derived input functions are shown in (**D**). Data are mean ± S.D Statistical comparison of values obtained at 15, 30 and 60 min after injection of [^18^F]FDS is reported with * *p* < 0.05, ** *p* < 0.01, n.s = nonsignificant.

**Figure 4 pharmaceutics-13-01752-f004:**
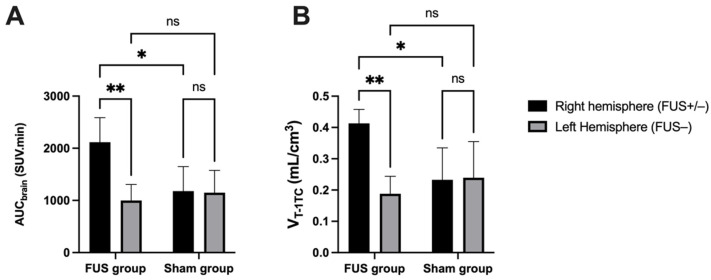
Comparison of [^18^F]FDS brain PET data. Ultrasound-induced BBB disruption (FUS+) was obtained in the right brain hemisphere of the FUS group (*n* = 5) while BBB in the contralateral brain hemisphere was intact (FUS−). The brain exposure of [^18^F]FDS was estimated by the area under the time-activity curve (AUC_brain_, in (**A**). Brain distribution was estimated by *V*_T_-_1TC_ (1-tissue compartment model, in (**B**) using an image-derived input function. Data were also compared with a sham group (no FUS, *n* = 3). Data are reported as mean ± S.D with * *p* < 0.05 and ** *p* < 0.01.

**Figure 5 pharmaceutics-13-01752-f005:**
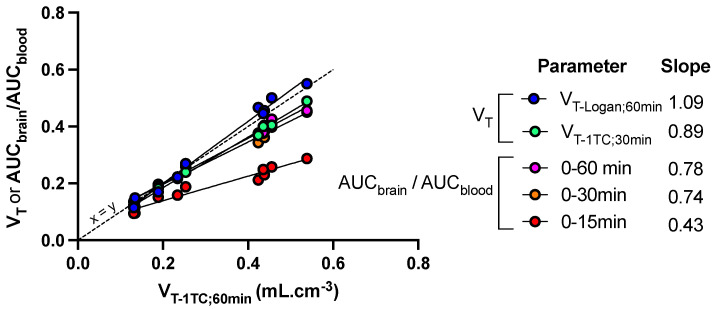
Sensitivity of kinetic parameters to describe the impact of FUS on the brain distribution of [^18^F]FDS in mice. Mice (*n* = 5) underwent blood-brain barrier disruption induced by focused ultrasound in the right hemisphere only. Kinetic modeling was performed to estimate the total brain distribution (*V*_T_) using the one-tissue compartment model (*V*_T-1TC_) estimated from 0 to 60 min after injection of [^18^F]FDS. *V*_T-1TC;0–60min_ values were correlated with *V*_T-1TC_ estimated from 0 to 30 min (V_T-1TC;0–30min_), *V*_T_ estimated using the Logan graphical method from 0 to 60 min (*V*_T-Logan;60min_) or the ratio of the time-activity curves (AUC) measured in brain regions and blood (AUC_brain_/AUC_blood_) from 0 to 15, 30 or 60 min. The slope of the linear correlation is reported.

## Data Availability

Data is contained within the article or Appendix A.

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
