# Peer review of "[^18^F]2-Fluoro-2-deoxy-sorbitol PET Imaging for Quantitative Monitoring of Enhanced Blood-Brain Barrier Permeability Induced by Focused Ultrasound"

_pharmaceutics, 2021, doi:10.3390/pharmaceutics13111752_

Round 1

Reviewer 1 Report

The manuscript is really well written, the results are thoroughly discussed and supported. 

I only found probably a typo at line 302. 

Author Response

The manuscript is really well written, the results are thoroughly discussed and supported. I only found probably a typo at line 302. 

We thank the reviewer for his/her comments. The typo has been corrected (“De” changed to “the”, now line 402).

Reviewer 2 Report

The manuscript entitle "[18F]2-fluoro-2-deoxy-sorbitol PET imaging for quantitative 2 monitoring of enhanced blood-brain barrier permeability 3 induced by focused ultrasound" highlights an interesting technique for measuring the transient Blood-Brain Barrier (BBB) permeability which if proven safe and effective can be a great diagnostic and possibly drug monitoring tool. Although, the use of focused ultrasound and microbubble sonication method has been studied for several years now, the authors of this manuscript introduced an interesting update that can enhance the utility of this method. While the approach and efforts are significant, I struggled to understand the of the methodologies use in this work for specifically measuring BBB disruption  and extravasation. Below are my major concerns:

  1. What is the rational for not washing out the BBB blood after the Evans blue and the [18F]FDS injections. How did you determine that is a true extravasation and not coming from the blood inside the BBB vessels. Also what is the rational for using Evans Blue! with all the disadvantages surrounding the use of EB and specifically what you stated in lines 60-62. Accordingly, Figure 2 is not clear without washing out the EB.
  2. Did you try to investigate the short and long-term impact of FUS on the BBB integrity. Did you consider measuring and quantifying the expression of the tight junction proteins and what is the impact of this on BBB major transporters.
  3. Introducing FUS induced BBB disruption as a method for treating neurological diseases such as Alzheimer's disease is a very sensitive topic especially with the large body of evidence supporting the importance of enhancing BBB integrity and function and how BBB breakdown can be an early marker in aging and different forms of dementia. Therefore, it is very important to specifically discuss the significance of the temporary use of FUS for a specific indication and it is very critical for the translational purposes to understand what is the long-term impact of this technique on the BBB integrity and function.
  4. Limitations of this study needs to be clearly addressed in the discussion.

Minor comments:
1. Several acronyms needs to be spelled out in the main text in their first appearance even if mentioned in the figures legends or the abstract. 

2. Introduction: First paragraph, consider discussing the role of BBB drug transporters in line 34 ( you can take a look at the following recent review (https://www.mdpi.com/1422-0067/22/7/3742)

3.Figure 3 add the significance (eg. P value and indicate with *) wherever possible.

4. line 135 add a "full stop" skull of 20% at 1.5 MHz [13]

5.Methods: using female NMRI nu/nu mice. talk about this model and What is the rational for using only using females and the low number of mice per group needs to be added as a limitation. 

6. Discuss in more details the translational value of this technique and what is the long term safety concerns in humans!

Author Response

We thank the reviewers for their constructive comments. We have addressed all points they have raised and changed the manuscript accordingly.

The manuscript entitle "[18F]2-fluoro-2-deoxy-sorbitol PET imaging for quantitative 2 monitoring of enhanced blood-brain barrier permeability 3 induced by focused ultrasound" highlights an interesting technique for measuring the transient Blood-Brain Barrier (BBB) permeability which if proven safe and effective can be a great diagnostic and possibly drug monitoring tool. Although, the use of focused ultrasound and microbubble sonication method has been studied for several years now, the authors of this manuscript introduced an interesting update that can enhance the utility of this method. While the approach and efforts are significant, I struggled to understand the of the methodologies use in this work for specifically measuring BBB disruption and extravasation. Below are my major concerns:

  1. What is the rational for not washing out the BBB blood after the Evans blue and the [18F]FDS injections. How did you determine that is a true extravasation and not coming from the blood inside the BBB vessels. Also what is the rational for using Evans Blue! with all the disadvantages surrounding the use of EB and specifically what you stated in lines 60-62. Accordingly, Figure 2 is not clear without washing out the EB.

We thank the reviewer for this comment. Evan’s blue provides a very simple, although poorly quantitative method to assess BBB disruption induced by focused ultrasound (FUS) ex vivo. Moreover, it can be easily combined with PET imaging in the same animal because EB extravasation can be assessed after PET acquisition. EB extravasation test has therefore been used as a positive control to visually check (on/off) and localize BBB disruption induced by FUS, which could be achieved without washing out. For reviewer information, perfusion of animals for washing-out of blood immediately after PET acquisition is technically possible but raises serious issues regarding radiation exposure of the staff.

Please see lines 141-143 and 189-195.

Several sentences were added to the manuscript to clarify that visual comparison of EB staining in the sonicated versus the non-sonicated brain in the same animal takes the circulating EB (inside de vessels) into account to assess BBB disruption. Please see lines 216-224.

  1. Did you try to investigate the short and long-term impact of FUS on the BBB integrity. Did you consider measuring and quantifying the expression of the tight junction proteins and what is the impact of this on BBB major transporters.

In the present study, only the short-term impact of FUS on BBB integrity has been investigated to validate [18F]FDS PET as a novel marker of BBB permeability. However, we think that the quantitative value of [18F]FDS PET may further improve our understanding of the short and long-term impact of FUS of BBB permeability as the next step of the present proof-of-concept study, as suggested by the reviewer. A sentence has been added to discuss the perspectives of the present work. Please see lines 359-365.

  1. Introducing FUS induced BBB disruption as a method for treating neurological diseases such as Alzheimer's disease is a very sensitive topic especially with the large body of evidence supporting the importance of enhancing BBB integrity and function and how BBB breakdown can be an early marker in aging and different forms of dementia. Therefore, it is very important to specifically discuss the significance of the temporary use of FUS for a specific indication and it is very critical for the translational purposes to understand what is the long-term impact of this technique on the BBB integrity and function.

We fully agree with this important comment. As paragraph has been added to mention strategies to restore BBB structure and function has a therapeutic target in neurological diseases. The counterintuitive “therapeutic” use of FUS has been moved to the discussion and is now explained with respect to the loss in BBB integrity and function observed in several CNS diseases. Please see lines 402-410.

  1. Limitations of this study needs to be clearly addressed in the discussion.

We now insist on the limitations of the study in the discussion. Please see lines 377-401.

Minor comments:
1. Several acronyms needs to be spelled out in the main text in their first appearance even if mentioned in the figures legends or the abstract. 

The acronyms are now defined the first time they appear in each of three sections: the abstract; the main text; the first figure, according to the guidelines of the journal.

  1. Introduction: First paragraph, consider discussing the role of BBB drug transporters in line 34 ( you can take a look at the following recent review (https://www.mdpi.com/1422-0067/22/7/3742)

Thank you for this suggestion. This interesting article is now cited in the introduction (line 34). Please see lines 387 and 408.

3.Figure 3 add the significance (eg. P value and indicate with *) wherever possible.

The P values have been added in Figure 3.

  1. line 135 add a "full stop" skull of 20% at 1.5 MHz [13]

This has been added accordingly. Please see line 129.

5.Methods: using female NMRI nu/nu mice. talk about this model and What is the rational for using only using females and the low number of mice per group needs to be added as a limitation. 

Selection of the mouse strain and gender is now clarified line 112-115. The potential impact of gender, strain, immunodeficiency and the low number of animals is now discussed as a limitation of the study line 392-401.

  1. Discuss in more details the translational value of this technique and what is the long term safety concerns in humans!

A sentence has been added to discuss the translational value of the technique and its limitations for use in humans. Please see lines 376-378.

Reviewer 3 Report

The manuscript by Hugon et al. describes the use of fluorine-18-radiolabeled fluorodeoxy-sorbitol (18F-FDS) in conjunction with PET imaging for the quantitative assessment of blood-brain barrier (BBB) opening, where the BBB of mice was opened by application of focused ultrasound (FUS) and injected microbubbles. The authors show that 18F-FDS readily enters the brain after FUS and can be quantitated by kinetic modeling. They conclude that this 18F-FDS-based method “provides a readily available, sensitive and reproducible marker of BBB permeability to non-invasively monitor the extent of BBB disruption induced by FUS in vivo.”

In all, this manuscript is an interesting report demonstrating that 18F-FDS can be used as a radiotracer for the assessment of BBB opening specifically by the method of FUS in mice. There are, however, two major issues that minimize the general impact of this report.

1. The novelty of the findings are not clearly stated, and the implied advantages of 18F-FDS over traditional 18F-FDG (fluorodeoxy-glucose) are not documented. An earlier study by others (Li et al, 2008; reference #32) entitled “Synthesis of 18F-FDS and its potential application in molecular imaging” already has introduced this radiotracer as a novel tool for PET imaging. In that study, the authors compared 18F-FDS side-by-side to conventional 18F-FDG and showed in mouse brain tumor models that 18F-FDS accumulated preferentially in tumor tissue, but not in normal brain tissue, thus providing better differentiation between normal and tumor tissue as compared to 18F-FDG. Yet, despite this apparent advantage published 13 years ago, 18F-FDS has not achieved widespread application in clinical imaging of brain tumors, which seems surprising. The authors should comment (in the Discussion section) why they expect that their application would be received more favorably than the earlier one.

In the current study by Hugon et al., it is now presented that 18F-FDS can enter normal brain tissue after BBB opening in a mouse model. However, 18F-FDS was not directly compared to 18F-FDG. It therefore remains unclear whether the hypothesized advantage of 18F-FDS over 18F-FDG indeed exists for purposes of non-invasive quantitation of BBB opening. This needs to be confirmed experimentally, where 18F-FDS is compared side-by-side to 18F-FDG under conditions of BBB opening.

2. The results are not well presented and explained, and overall are sketchy, which makes it difficult to follow this report. There are several major and minor issues that require revision:

a. Text from Line 209 to 219 is a repeated paragraph and needs to be deleted.

b. Figure 1 is not mentioned in the Results section. Although this figure is mentioned in the Introduction, the Results section opens with 18F-FDS production and should refer to that figure. The Legend to Figure 1 is too sketchy. More details should be added. NaBH4 needs to be defined and explained. All reaction products should be shown (what happens to Na and B after the reaction; i.e., provide complete reaction equation). Check Line 187 “which Rf in the same condition” (?).

c. Throughout the text, the figures should be presented in ascending order and the results introduced in order of the figures. Right now, the Results section goes back and forth among the figures and concludes with Figure 2D (from among 5 figures). Figure 2A, B, C is not mentioned in the text, and neither is Figure 3B. Each panel of each figure should be worth at least one descriptive sentence in the main text.

d. In Figure 4, the asterisks representing p-values are missing. Legend does not mention A, B.

e. The choice of animals should be justified. Why did the authors use NMRI nu/nu mice, which are immunocompromised? Why not “regular” mice?

f. Please use consistency. No need to capitalize “silica”, “sham” or “blue” (as in Evan’s blue). Use either Figure or Fig., but not both. Add spaces where needed. Adjust superscript “18F” difference between main text and figure legends. Define AP-HP, CNRS, BAOBAB, etc.

g. Please proofread carefully.

Author Response

The manuscript by Hugon et al. describes the use of fluorine-18-radiolabeled fluorodeoxy-sorbitol (18F-FDS) in conjunction with PET imaging for the quantitative assessment of blood-brain barrier (BBB) opening, where the BBB of mice was opened by application of focused ultrasound (FUS) and injected microbubbles. The authors show that 18F-FDS readily enters the brain after FUS and can be quantitated by kinetic modeling. They conclude that this 18F-FDS-based method “provides a readily available, sensitive and reproducible marker of BBB permeability to non-invasively monitor the extent of BBB disruption induced by FUS in vivo.”

 In all, this manuscript is an interesting report demonstrating that 18F-FDS can be used as a radiotracer for the assessment of BBB opening specifically by the method of FUS in mice. There are, however, two major issues that minimize the general impact of this report.

  1. The novelty of the findings are not clearly stated, and the implied advantages of 18F-FDS over traditional 18F-FDG (fluorodeoxy-glucose) are not documented. An earlier study by others (Li et al, 2008; reference #32) entitled “Synthesis of 18F-FDS and its potential application in molecular imaging” already has introduced this radiotracer as a novel tool for PET imaging. In that study, the authors compared 18F-FDS side-by-side to conventional 18F-FDG and showed in mouse brain tumor models that 18F-FDS accumulated preferentially in tumor tissue, but not in normal brain tissue, thus providing better differentiation between normal and tumor tissue as compared to 18F-FDG. Yet, despite this apparent advantage published 13 years ago, 18F-FDS has not achieved widespread application in clinical imaging of brain tumors, which seems surprising. The authors should comment (in the Discussion section) why they expect that their application would be received more favorably than the earlier one.

We thank the reviewer for this important comment. The article by Li et al. 2008 cited by the reviewer addressed the negligible uptake of 18F-FDS by cancer cells in vitro but did not provide mechanistic explanation regarding the enhanced brain uptake of 18F-FDS in an orthotopic animal model of brain tumor. Our results, obtained using a method for controlled BBB disruption, reveals the high and so far unreported sensitivity of 18F-FDS to BBB integrity. In can therefore be hypothesized that enhanced PET signal observed by Li et al.  in the tumor area may be linked to local BBB disruption. Our sentence was rephrased to take reviewers’ comment into account in the revised manuscript. Please see lines 295-302.

In the current study by Hugon et al., it is now presented that 18F-FDS can enter normal brain tissue after BBB opening in a mouse model. However, 18F-FDS was not directly compared to 18F-FDG. It therefore remains unclear whether the hypothesized advantage of 18F-FDS over 18F-FDG indeed exists for purposes of non-invasive quantitation of BBB opening. This needs to be confirmed experimentally, where 18F-FDS is compared side-by-side to 18F-FDG under conditions of BBB opening.

18F-FDG undergoes facilitated transport across the BBB so that high baseline uptake of 18F-FDG is observed across the intact BBB. The impact of FUS-induced BBB disruption on the brain uptake of 18F-FDG has already been published. This is now clearly reported lines 349-353.

  1. The results are not well presented and explained, and overall are sketchy, which makes it difficult to follow this report. There are several major and minor issues that require revision:
  2. Text from Line 209 to 219 is a repeated paragraph and needs to be deleted.

This was a copy-paste error. Thank you, the paragraph has been deleted.

  1. Figure 1 is not mentioned in the Results section. Although this figure is mentioned in the Introduction, the Results section opens with 18F-FDS production and should refer to that figure.

Figure 1 is now cited in the result section line 184.

The Legend to Figure 1 is too sketchy. More details should be added. NaBH4 needs to be defined and explained. All reaction products should be shown (what happens to Na and B after the reaction; i.e., provide complete reaction equation).

Legend to Figure 1 has been improved accordingly. Please see lines 106-108.

Check Line 187 “which Rf in the same condition” (?).

The sentence has been rephrased lines 186-188.

  1. Throughout the text, the figures should be presented in ascending order and the results introduced in order of the figures. Right now, the Results section goes back and forth among the figures and concludes with Figure 2D (from among 5 figures). Figure 2A, B, C is not mentioned in the text, and neither is Figure 3B. Each panel of each figure should be worth at least one descriptive sentence in the main text.

 Figures are now presented and cited in the correct order in the manuscript.

  1. In Figure 4, the asterisks representing p-values are missing. Legend does not mention A, B.

Asterisks have been added in Figure 4 of the revised manuscript, thank you.

  1. The choice of animals should be justified. Why did the authors use NMRI nu/nu mice, which are immunocompromised? Why not “regular” mice?

 A sentence has been added to justify the use of NMRI nu/nu mice in this project. Please see lines 112-115 and in the discussion section lines 392-401.

  1. Please use consistency. No need to capitalize “silica”, “sham” or “blue” (as in Evan’s blue). Use either Figure or Fig., but not both. Add spaces where needed. Adjust superscript “18F” difference between main text and figure legends. Define AP-HP, CNRS, BAOBAB, etc.

We have corrected the manuscript accordingly. We have been asked by our institution not to define the affiliations AP-HP, CNRS and BOABAB.

  1. Please proofread carefully.

The revised manuscript has been double-checked. We thank the reviewer for his/her efforts to improve the manuscript.

Round 2

Reviewer 2 Report

The manuscript quality improved and the authors responded to all the comments.

Author Response

We thank the reviewer for his/her comments.

Reviewer 3 Report

The authors responded to most comments, but did not apply all of their revisions.

This reviewer asked for revisions to Figure 4, and the authors responded that they now revised Figure 4 accordingly. However, the revised version of the manuscript does not show the requested changes to Figure 4. (1) The figure legend defines the p-values with one or two asterisks, and these asterisks need to be added to the figure. Please add asterisks to the figure. (2) The figure displays two parts, A and B. But the figure legend does not mention A or B. Please define A and B in the figure legend.

Reviewer 2 had requested that the authors "add the significance (eg P value and indicate with *)" to Figure 3, and the authors responded that they added the p-values in Figure 3. However, the revised version of the manuscript does not show these changes to Figure 3, i.e., there are no p-values in this figure. Please add p-values to the figure.

Author Response

We thank the reviewer for his/her comments

The authors responded to most comments, but did not apply all of their revisions.

- This reviewer asked for revisions to Figure 4, and the authors responded that they now revised Figure 4 accordingly. However, the revised version of the manuscript does not show the requested changes to Figure 4.

(1) The figure legend defines the p-values with one or two asterisks, and these asterisks need to be added to the figure. Please add asterisks to the figure.

(2) The figure displays two parts, A and B. But the figure legend does not mention A or B. Please define A and B in the figure legend.

Response: Asterisks disappeared during the conversion of the file. Figure 4, including figure legend, has now been changed accordingly. Thank you.

- Reviewer 2 had requested that the authors "add the significance (eg P value and indicate with *)" to Figure 3, and the authors responded that they added the p-values in Figure 3. However, the revised version of the manuscript does not show these changes to Figure 3, i.e., there are no p-values in this figure. Please add p-values to the figure.

Response:  Sorry, it is a misunderstanding on our part. We have now added p-values in each panel of Figure 3 to highlight statistical differences at selected time-points. Corresponding information have been added in the material and methods section lines 179-181, in the result section lines 229-237, and in the legend of Figure 3 lines 245-246.

Round 3

Reviewer 3 Report

Authors addressed most comments. Manuscript is okay now.